# Antioxidant and Lipid-Lowering Effects of Buriti Oil (*Mauritia flexuosa* L.) Administered to Iron-Overloaded Rats

**DOI:** 10.3390/molecules28062585

**Published:** 2023-03-13

**Authors:** Jailane de Souza Aquino, Kamila Sabino Batista, Gabriel Araujo-Silva, Darlan Coutinho dos Santos, Naira Josele Neves de Brito, Jorge A. López, João Andrade da Silva, Maria das Graças Almeida, Carla Guzmán Pincheira, Marciane Magnani, Débora C. Nepomuceno de Pontes Pessoa, Tânia L. Montenegro Stamford

**Affiliations:** 1Experimental Nutrition Laboratory, Department of Nutrition, Federal University of Paraíba (UFPB), João Pessoa 58051-900, PB, Brazil; 2Organic Chemistry and Biochemistry Laboratory, State University of Amapá (UEAP), Macapá 68900-070, AP, Brazil; 3Experimental Nutrition Research Group, Vive Sano University Institute (IUVS), São Paulo 04304-000, SP, Brazil; 4Health Science Postgraduate Department, University of Cuiabá (UNIC), Sinop 78550-100, MT, Brazil; 5Department of Food Technology, Center for Technology and Regional Development, Federal University of Paraíba (UFPB), João Pessoa 58051-900, PB, Brazil; 6Department of Clinical and Toxicological Analysis, Federal University of Rio Grande do Norte (UFRN), Natal 59078-970, RN, Brazil; 7College of Health Care Sciences, Concepción Campus, San Sebastian University, Concepción 4030000, Chile; 8Laboratory of Microbial Processes in Food, Department of Food Engineering, Federal University of Paraíba (UFPB), João Pessoa 58051-900, PB, Brazil; 9Department of Nutrition, Federal University of Pernambuco (UFPE), Recife 50670-901, PE, Brazil

**Keywords:** antioxidant activity, carotenoids, fatty acids, tocopherol, iron, vegetable oils

## Abstract

The indiscriminate use of oral ferrous sulfate (FeSO_4_) doses induces significant oxidative damage to health. However, carotene-rich foods such as buriti oil can help the endogenous antioxidant defense and still maintain other body functions. This study aimed to assess the effects of buriti oil intake in iron-overloaded rats by FeSO_4_ administration. Buriti oil has β-carotene (787.05 mg/kg), α-tocopherol (689.02 mg/kg), and a predominance of monounsaturated fatty acids (91.30 g/100 g). Wistar rats (*n* = 32) were subdivided into two control groups that were fed a diet containing either soybean or buriti oil; and two groups which received a high daily oral dose of FeSO_4_ (60 mg/kg body weight) and fed a diet containing either soybean (SFe) or buriti oil (Bfe). The somatic and hematological parameters, serum lipids, superoxide dismutase (SOD), and glutathione peroxidase (GPx) were determined after 17 days of iron overload. Somatic parameters were similar among groups. BFe showed a decrease in low-density lipoprotein (38.43%) and hemoglobin (7.51%); an increase in monocytes (50.98%), SOD activity in serum (87.16%), and liver (645.50%) hepatic GPx (1017.82%); and maintained serum GPx compared to SFe. Buriti oil showed systemic and hepatic antioxidant protection in iron-overloaded rats, which may be related to its high carotenoid, tocopherol, and fatty acid profile.

## 1. Introduction

Iron plays a key role in various cellular processes (e.g., oxygen transport, cell proliferation, and catalytic reactions) [1]. Despite the important metabolic role of iron in the body, iron overload is responsible for several disorders, whether due to hereditary hemochromatosis or to indiscriminate supplementation, especially in children, pregnant women, nursing mothers, and athletes [2,3,4]. Iron overload can cause anemia and even chronic intoxication due to its accumulation in various organs, promoting oxidative stress and inflammatory responses and, consequently, damage at the cellular level [5,6].

Thus, the organism’s iron balance must be maintained by meticulous regulation of its intestinal absorption release from macrophages to satisfy metabolic or functional demands and prevent deleterious effects due to iron deficiency or excess [7]. An overloaded iron condition is an important oxidative damage inducer, favoring reactive oxygen species (ROS) production by the Fenton reaction [8]. ROS are continuously produced by normal cell metabolism, causing structural and molecular damage, contributing to aging and diseases (e.g., cancer and diabetes) [9]. At this point, iron overload may increase cellular oxidative stress [10,11,12], and its hepatic accumulation modifies the liver enzyme gene expression involved in lipid metabolism, increasing serum and tissue cholesterol [13,14].

On the other hand, this oxidative damage in animals is minimized by an efficient antioxidant system, such as enzymes (e.g., superoxide dismutase and glutathione peroxidase), and low-molecular-weight compounds such as glutathione and ascorbic acid, or nonpolar compounds such as carotenoids to scavenge ROS and defend cells and macromolecules from oxidative damage. Although ROS act as secondary messengers, a redox imbalance is established under certain metabolic, physiological, and pathological conditions, in which the antioxidant defense systems are insufficient to scavenge oxidative compounds [12,15].

So far, drugs for the treatment of excess iron cause many side effects, whether it be iron overload generated by storage-related diseases or excessive iron supplementation. Chelation therapy is the only method applied to iron excretion despite its adverse effects [16], which in turn has stimulated studies with vegetable foods, medicinal plants, and their phytocomposition, rich in antioxidant compounds to control the redox imbalance caused by iron [17,18,19,20,21]. Thus, natural products with significant content in β-carotene, tocopherols, and polyphenols are efficient sources of exogenous antioxidants in regulating iron overload and are potential candidates for developing herbal medicines and/or functional foods [15,22].

In this context, the oil extracted from the buriti fruit pulp (*Mauritia flexuosa* L.f.) is a remarkable natural product due to its high carotenoid, tocopherol, and monounsaturated fatty acid (MUFA) content [23,24,25,26,27]. The growing interest in buriti oil is due to biological activities demonstrated in silico, in vitro, and in vivo [28]. Constituents of buriti oil have been shown to act as major virus SARS-Coronavirus peptidase inhibitors [29] and have shown antimicrobial activity [30]. Furthermore, buriti oil modulated physical parameters and reflex maturation of the offspring of dams fed with this oil; improved the lipid profile; and increased serum, hepatic retinol, and tocopherol in young rats [23,26].

Based on this, the purpose of this study was to assess the buriti oil intake effect in iron-overloaded rats by evaluating somatic parameters, serum lipid and hematological profile, and superoxide dismutase (SOD) and glutathione peroxidase (GPx) activities.

## 2. Results

### 2.1. Oil Characterization

The characterization of the oils and the antioxidant capacity are presented in Table 1 and Appendix A, respectively. Buriti oil showed important levels of β-carotene and tocopherol, in addition to a higher concentration of monounsaturated fatty acids (MUFAs) such as oleic and palmitoleic acids, and a lower concentration of saturated (SFA) and polyunsaturated (PUFA) fatty acids compared to soybean oil (*p* ≤ 0.001).

### 2.2. Evaluation of Somatic Parameters and Food Intake

Orogastric administration of a high FeSO_4_ dose in the SFe rat group resulted in a lower dietary intake compared to the SC and BFe groups (*p* ≤ 0.05) (Figure 1A). However, no significant differences (*p* > 0.05) regarding body mass gain (Figure 1B) and somatic parameters were observed among the groups (Table 2).

### 2.3. Effect of Buriti Oil on Lipid Profile, Aminotransferases, and Hematological Parameters

Iron overload induced damage to the lipid profile, such as a significant increase in TAG (Figure 2A), VLDL (Figure 2C), and aminotransferases concentrations (Figure 2F,G) being observed in the SFe and BFe groups in comparison to the SC and BC, respectively (*p* ≤ 0.05). This result can be supported by a strong and significant positive correlation between TAG and VLDL values (r = 0.84; *p* ≤ 0.001) (Figure 3 and Appendix A).

The SFe group displayed the highest LDL value (*p* ≤ 0.05) (Figure 2D) compared to the control group (SC) (*p* ≤ 0.05); however, buriti oil consumption reversed the increase in LDL caused by iron overload in the BFe group (Figure 2D). No significant difference was determined in the TC (Figure 2B) and HDL values (Figure 2E). A moderate positive correlation between TC and body weight (r = 0.66; *p* ≤ 0.001) was observed (Figure 3 and Appendix A), and an inverse correlation between C 18:0 acid (or stearic acid) and LDL seric (r = −0.66), and ALT (r = −0.62) and AST (r = −0.63) was observed (*p* ≤ 0.001). Furthermore, there is a positive correlation of LDL values with ALT (r = 0.70) and AST (r = 0.71) values (*p* ≤ 0.001) (Figure 3 and Appendix A).

The administration of iron overload altered some parameters in the hematological profile (Figure 4), regardless of the type of oil administered, such as hematocrit (Figure 4B), WBCs (Figure 4E), and lymphocytes (Figure 4F). Furthermore, WBCs and lymphocytes had a strong and significant positive correlation (r = 0.77; *p* ≤ 0.001) (Figure 3 and Appendix A). On the other hand, buriti oil consumption reduced hemoglobin (Figure 4C) and increased monocyte counts (Figure 4G) in BFe rats compared to SFe rats, although no significant difference was observed among control groups (SC and BC) (*p* > 0.05) (Figure 4G). There was no significant difference regarding red blood cells and platelets among any of the groups (Figure 4A and Figure 4D, respectively).

### 2.4. Effect of Buriti Oil on Antioxidant Enzyme Activity in Serum and Liver

The administration of the high-FeSO_4_ dose decreased serum (Figure 5A) and liver SOD activity (Figure 5B) in the SFe group (*p* ≤ 0.05). On the other hand, intake of the diet containing buriti oil increased SOD enzymatic activity in the BC and BFe groups (*p* ≤ 0.05) (Figure 5A,B). The BFe group maintained a similar GPx serum activity level to the other groups (*p* > 0.05), while the SFe group exhibited the lowest GPx compared to the SC group (*p* ≤ 0.05) (Figure 5C). An expressive increase in hepatic GPx activity was observed in rat groups fed a diet containing buriti oil (BC and BFe) (*p* ≤ 0.05) (Figure 5D). Furthermore, hepatic SOD (r = 0.61) and serum GPx (r = 0.62) were positively correlated with blood VLDL levels (*p* ≤ 0.001) (Figure 3 and Appendix A).

## 3. Discussion

Our study demonstrated that buriti oil has important antioxidant activity in vitro, as well as bioactive compounds such as β-carotene, α-tocopherol, and MUFAs such as palmitoleic and oleic acids. Previous studies have demonstrated a similar nutritional characterization of this oil, which has a predominance of monounsaturated fatty acids, followed by saturated and polyunsaturated fatty acids [23,31]. Buriti oil has a higher concentration of β-carotene, α-tocopherol, and oleic acid compared to oils extracted from the mesocarp of other Brazilian palm trees (*Arecaceae*), such as bacaba (*Oenocarpus bacaba*), inajá (*Maximiliana maripa*), pupunha (*Bactris gasipaes*), tucumã (*Astrocaryum vulgare*), and buritirana (*Mauritiella armata)* [32,33,34,35].

Our results demonstrate that iron overload administration in rats increased TAG, VLDL, and LDL values, which were associated with decreased antioxidant defense mechanisms, mainly hepatic and serum antioxidant enzymes (SOD and GPx). Otherwise, we demonstrated that buriti oil intake attenuated iron-overload-induced changes in hematological parameters, lipid profile, and serum and hepatic antioxidant status in BFe rats.

Regarding dietary intake, the SFe group consumed less diet than the other groups, probably due to the ferrous sulfate taste interference on the rat gustatory sensitivity to the standard diet [36]. Although the BFe rats were also treated with FeSO_4_, no decrease in food consumption was observed, which may indicate that the buriti oil presence in the diet promoted a palatable effect [37]. However, we demonstrated that the consumption of diets containing different lipid sources (buriti or soybean oil) did not change the somatic parameters in young adult rats, which was also observed by Aquino et al. [23,38] in rats in childhood and adolescence phases who were fed diets added with either soybean or buriti oil for 28 consecutive days. Iron overload also did not influence these parameters, probably due to the short time of high-FeSO_4_ dose administration. In addition, the consumption of a high-fat diet associated with iron overload seems to enhance changes compatible with the metabolic syndrome in mice [39], which in the long term can cause somatic changes. However, the diets administered to the rats in the present study are isolipidic, and they followed dietary recommendations for rodents, which may have also contributed to these results.

Iron overload in the present study caused deleterious oxidative effects on lipid metabolism, increasing TAG and VLDL levels in the SFe and BFe groups. In addition, a significant positive correlation between these parameters was observed in our study. This is interesting due to the VLDL lipoprotein transporting triglycerides to the bloodstream and organs, which can promote the development of vascular lesions [40]. On the other hand, buriti oil reversed the increase in LDL levels in the BFe rats caused by iron overload, which was not observed in the SFe rats. Although stearic acid (C 18:0) showed an inverse correlation with LDL levels, as confirmed by previous studies [41,42], soybean and buriti oils showed similar concentrations of this fatty acid, which may indicate the effect of other compounds present in buriti oil, such as oleic acid, β-carotene, and α-tocopherol, in reducing LDL levels.

Studies have shown that unsaturated fatty acid intake can act as a protective agent against physiological injuries due to anti-inflammatory and antioxidant effects [43,44,45]. MUFAs can interfere with lipid metabolism through some mechanisms, which can decrease the sterol storage, the regulation of cholesterol synthesis, cellular absorption of LDL, and fat oxidation [46]. Oleic acid is the majority fatty acid in buriti oil, and this MUFA plays an important role in the peroxisome-proliferator-activated receptor-α (PPAR-α) upregulation, which in turn modulates the uptake, transport, and oxidation of fatty acids in rats induced to iron overload via fatty acid transporters [47].

Furthermore, the high carotenoid and tocopherol contents in buriti oil may be responsible for minimizing lipoprotein oxidation and, consequently, lowering serum LDL levels. Studies have shown the β-carotene- and α-tocopherol-modulating roles in pathways related to oxidative stress and inflammation, indicating that its dietary supplementation improved the serum lipid profile in rats fed a cholesterol-rich diet [23,48,49,50,51,52].

Although no histopathological analysis of the liver and other organs was performed, serum aminotransferase levels (ALT and AST) are reliable indicators of the functional or structural alteration of liver cells [53]. ALT and AST are widely used as markers of liver injury and inflammation in clinical practice [54]. In this concern, iron overload caused an increase in the ALT and AST concentrations in the SFe and BFe groups compared to the SC and BC groups, respectively. Nevertheless, it was evidenced that buriti oil consumption attenuated the elevation of both aminotransferases in BFe rats fed buriti oil compared to animals in the SFe group.

The current treatment of iron overload includes metal chelators, which can cause adverse effects. For example, deferoxamine, deferiprone, and deferasirox increase the risk of agranulocytosis and neutropenia, risk of gastrointestinal disturbances, and hepatic failure. These adverse effects of available iron chelators highlight the need for alternative pharmacological interventions [55], such as buriti oil, which showed protective effects against damage induced to iron overload in rats.

We demonstrated that the buriti oil intake attenuated the toxicity promoted in rats after iron-overload treatment, according to an evaluation of the hematological parameter data (erythrocytes, leukocytes, and platelet counts). Elevated hematocrit and hemoglobin levels exhibited by the SFe group compared to the SC group may be considered indicative of injury from a high-FeSO_4_ dose. Nevertheless, the BFe group displayed a decrease in hemoglobin content, probably due to a regulatory buriti oil effect on hemoglobin synthesis [6]. The buriti oil β-carotene content may have promoted maintenance of hemoglobin levels. β-carotene regulates iron metabolism, and its anti-inflammatory effect upregulates erythropoietin expression and iron mobilization for erythropoiesis [15,56].

Concerning the action mode of buriti oil, evidence indicates that both β-carotene and oleic acid interfere with iron absorption and metabolism. Buriti oil displays a very complex bioactive chemical composition, with β-carotene and oleic acid as major components, which may explain, due to a possible synergistic action, the iron absorption attenuation and metabolism in FeSO4-treated animals. At this point, studies with oils and other natural products have reported their effectiveness as a synergistic interaction result, in which it is difficult to determine the compound responsible for a biological effect since the chemical complexity represents a great challenge for its identification. Indeed, studies have indicated the behavior of a mixture as a general activity due to the synergistic effect related to the structural interaction of compounds belonging to different classes [57,58].

Iron overload had no effect on platelet counts, although it did impact leukocyte counts in the SFe group. Furthermore, WBCs and lymphocytes had a significant positive correlation, with the increase in lymphocytes (the WBC subtype) occurring in response to the injury caused by excess FeSO_4_ administered to the rats, since lymphocytes are a first-line protective barrier against the deleterious effects of excess iron [59].

Iron metabolism regulation occurs at systemic and cellular levels by several coordinated mechanisms involving erythrocytes, monocytes/macrophages, enterocytes, and hepatocytes through signaling triggered by the hepcidin–ferroportin interaction [6,60]. Thus, the results of this study indicate that the FeSO_4_ overload administration may have disrupted the iron metabolism in rats, promoting an inflammatory response detected by a significant increase in lymphocyte and granulocyte counts. The decrease in the monocytes (macrophage precursors) in the SFe rat group can be associated with this cell migration to several organs as an inflammatory response to oxidative stress induced by iron overload in the animal body [60,61].

Excess iron content in the cell is harmful because it is involved in redox reactions, which induce oxidative stress. Iron is involved in the Fenton reaction by producing highly reactive hydroxyl radicals from hydrogen peroxide. The role of oxidative stress in the pathogenesis and progression of other diseases, including cardiovascular disorders, is well established. Otherwise, carotenoids (mainly β-carotene) are the main nonpolar antioxidants in cells, presenting in buriti oil in a relevant quantity, and showing in vivo chelator potential, antioxidant activity, and enzymatic-induced activity [62,63,64].

In our study, the elevation in oxidative stress in chronically iron-overloaded rats was associated with possible depletion of nonenzymatic as well as enzymatic (superoxide dismutase and glutathione peroxidase) antioxidants in the liver and blood. However, buriti oil administration significantly restored the activities of enzymatic antioxidants.

Concerning the antioxidant enzyme, the SOD and GPx activity decrease in the SFe group indicates the deleterious effect of a high-FeSO_4_ dose administration. However, the results from the group fed a diet with buriti oil (BFe) suggest that the oil chemical composition was efficient to minimize the deleterious effect of the high-FeSO_4_ dose on antioxidant enzyme activity. This result agrees with studies in which GPx and SOD activity evaluation in animals and humans with a β-carotene- or α-tocopherol-supplemented diet indicated the expression and activity of these enzymes, probably because of the compensatory defense of oxidative stress [64,65,66,67].

The high contents of β-carotene and α-tocopherol quantified in buriti oil contribute to the significant in vitro antioxidant capacity, which in turn can attenuate oxidative damage with consequent tissue recovery and biomolecules. β-carotene can quench singlet oxygen, α-tocopherol reduces mitochondrial hydrogen peroxide release rate, and both react as ROS scavengers in a nonenzymatic manner to prevent or decrease oxidative damage to biomolecules and cells [50,65,66,67,68]. Furthermore, a previous study demonstrated the synergism action among oleic acid and α-tocopherol to prevent oxidative stress [47].

Regarding the liver and blood-serum antioxidant enzyme activity significant differences, this event may occur because the liver is affected earlier by changes in diet and iron overload compared to enzymes analyzed at the systemic level. Therefore, the differences observed in the present study between liver and serum GPx and SOD activities may be related to specific dietary changes and liver lipid composition. This is supported by studies suggesting that antioxidant enzyme patterns in rats fed different dietary lipid sources are tissue-specific, meaning that the fat type in the diet can modify the responses to oxidative stress in specific organs, such as the liver [69].

Furthermore, the hepatic microsomal fatty acid composition may also play a role in the oxidative stress response since different fat types have been shown to affect this lipid composition [70]. A previous study has evidenced significantly lower hepatic levels of SOD, catalase, and GPx and SOD activities in nonalcoholic fatty liver disease compared to serum levels of these enzymes, respectively [71]. These results suggest a particular liver tissue vulnerability to oxidative damage and the importance of monitoring serum and liver antioxidant enzyme activity to obtain a more comprehensive assessment of oxidative stress. Overall, these studies highlight the importance of considering tissue-specific differences in antioxidant enzyme activity and possible implications for assessing pathogenesis and treatments [70].

It is noteworthy that the positive correlation between hepatic GPx and serum SOD values with the VLDL values is a defense mechanism against the oxidation of this lipoprotein, since a high-FeSO_4_-dose treatment induces liver oxidative damage and, consequently, lipid dysregulation. Iron-mediated lipid peroxidation generates ROS as well as oxidized lipoproteins, such as VLDL, an LDL precursor, promoting inflammation and foam cell formation, which in turn are factors associated with atherosclerosis [6,72].

The glutathione peroxidase family (GPx 1, 2, 3, and 4) is broadly distributed in different mammalian tissues. One of these (GPx4) binds to membranes and is responsible for nonapoptotic cell death regulation or ferroptosis due to excess iron exposure [73,74]. Therefore, the marked hepatic GPx depletion in the SFe rat group is indicative of the decrease or failure in the antioxidant enzyme defense system and the ferroptosis activation. Overall, the results show the attenuation of the metabolic effects caused by iron overload in rats fed a buriti oil diet. Notwithstanding, further studies are required regarding changes in antioxidant defense and the immune system, as well as focusing on liver histological evaluations to clarify other mechanisms involved and the progression of damage caused by iron overload.

## 4. Materials and Methods

### 4.1. Materials

Ripe buriti fruits (*Mauritia flexuosa* L.f.) were purchased at a popular market in Picos (PI, Brazil) for oil extraction and were identified and deposited (Nº 30567) by the Herbarium Graziela Barroso of the Universidade Federal do Piauí (UFPI). The diet ingredients were acquired from Rhoster^®^ (São Paulo, SP, Brazil), and refined soybean oil (Soya^®^, São Paulo, Brazil) was acquired in local markets (João Pessoa, PB, Brazil).

SOD and GPx activities were determined by RANSOD and RANSEL diagnostic kits (Randox Laboratories, County Antrim, UK), respectively. Diagnostic kits to assess serum lipid profile (triacylglycerides (TAG), total cholesterol (TC), high-density lipoprotein (HDL), and low-density lipoprotein (LDL)) were purchased from Labtest Reagents (Belo Horizonte, MG, Brazil). Fluorophore membranes (0.5 μm) were purchased from Millipore (Billerica, MA, USA), while HPLC-grade solvents and other analytical-grade chemicals were supplied by Merck (Darmstadt, Germany).

### 4.2. Oil Samples

Buriti oil was extracted from buriti pulp using a manual procedure. First, fruit pulp samples were suspended in distilled water, heated at 60 °C, and stirred for 30 min to extract the crude oil. Then, the crude oil was used in the refining process, followed by neutralization, washing, degumming, and drying steps, according to Aquino et al. [31].

The oil was neutralized with 5.0% sodium hydroxide solution at 12%, concerning the mass of oil, at a temperature of 50 °C, under stirring for 30 min. Subsequently, the oil was centrifuged at 5000 rpm and then transferred to a separatory funnel where successive washes were performed. The washes were carried out with water at room temperature alternating with water at a temperature of 90–95 °C at intervals of 30 min each, stirring the contents of the funnel manually. At each washing step, the water was discarded and evaluated with acid-base indicators (phenolphthalein and bromothymol blue) to detect any trace of sodium hydroxide used in the neutralization. The washes were completed when alkalinity was no longer detected in the discarded water. Then, the oil was dried in a rotary evaporator at 60 °C, with vacuum pressure for 20 min, under slight agitation and cooled to room temperature [31]. After the refining process, buriti oil was analyzed for the following parameters: free fatty acids (0.29 ± 0.03%), acidity index (0.22 ± 0.03%), and peroxide index (6.89 ± 0.78 mEq/kg), which demonstrated that the oil of refined buriti is within normal standards for consumption [31]. The extracted oil was stored at 7–10 °C in amber glass bottles for subsequent experiments.

### 4.3. Oil Chemical Characterization

The buriti oil and soybean oil were analyzed in triplicate by chromatographic methods to determine their respective fatty acids, as well as the β-carotene and α-tocopherol contents, according to Aquino et al. [23] (Table 1).

Fatty acid methyl esters (FAMEs) were obtained according to the methodology described by Hartman and Lago [69]. The FAMEs were separated, identified, and quantified on a gas chromatograph (Varian 430, Walnut Creek, USA), coupled to a flame ionization detector and an SPTM -2560 capillary column (100 m × 0.25 mm and film thickness of 0.20 μm, Supelco, Bellefonte, PA, USA). Helium (1 mL/min) was used as a carrier gas, and hydrogen (30 mL/min) and synthetic air (300 mL/min) were used as auxiliary gases. The temperatures of the split:splitless injector and the detector were maintained at 240 °C and 250 °C, respectively, with an injection volume of 1.0 μL. The initial oven temperature was 100 °C, which was then increased by 2.5 °C/min to 245 °C and maintained for 30 min. The FAME retention times of each oil were compared with a mixture of standards containing 19 methyl esters (ME19–Kit, Fatty Acid Methyl Esters C4-C24, Supelco, Bellefonte, USA) to identify the fatty acids, with results quantified by normalization areas of methyl esters. The chromatograms were recorded in the GalaxieTM Chromatography Data System software program (Varian, Palo Alto, CA, USA).

The β-carotene and α-tocopherol analyses were performed by HPLC (Shimadzu HPLC class 10) coupled with a diode detector (DAD) and a LiChrospher 100 RP C18 column (125 cm × 4 mm, 5 µm particle stationary phase, Merck, Darmstadt, Germany).

For β-carotene, the oil sample was cold saponified with a 10% KOH methanolic solution and the carotenoids extracted with petroleum ether. After washing to neutral pH, the extract solution volume was adjusted and filtered through PTFE 0.45 µm membrane to quantify the β-carotene content. A 200 µL sample was injected manually using methanol:chloroform (95:5) as a mobile phase, at a flow rate of 1 mL/min, monitored at 450 nm, with 13 min of retention time [23].

Next, 0.1 g of each oil was weighed, followed by dilution in 2 mL of 2-propanol, and filtration through 0.45 um PTFE filters for injection into the chromatograph for the tocopherol analysis. A Si 60 normal phase column of 125 × 4 mm internal diameter with 5 µm particles was used, with n-hexane:ethyl acetate:acetic acid (97.6:1.8:0.6, v/v/v) as mobile phase and a flow rate of 1.5 mL/min. Quantification of tocopherol was performed using wavelengths of 294 nm for excitation and 326 nm for emission, with external standardization and 3 min of retention time [23].

The buriti oil and soybean oil in vitro antioxidant potential was evaluated by the DPPH assay [23] (Appendix A), expressing the results as the percentage (%) of free radical scavenging and IC_50_ value as the oil concentration necessary to promote 50% of DPPH scavenging.

### 4.4. Animals, Diet, and Induction of Iron Overload

The study was carried in compliance with ARRIVE guidelines (Animal Research: Reporting of In Vivo Experiments) [75]. All animal procedures performed with prior approval by the UFPE Ethical Committee of Animal Research (protocol 23076.001000.2010-29).

Thirty-two (32) male Wistar rats (349 ± 15 g) were kept in cages at 22 ± 1 °C on a 12 h light/dark cycle (light on 07:00 p.m.) and at a relative humidity of 50–55%. Rats were randomized and allocated into four groups: two control groups fed diet containing soybean (SC, *n* = 8) or buriti oil (BC, *n* = 8) and daily saline solution via gavage; and two groups fed diet containing soybean (SFe, *n* = 8) or buriti oil (BFe, *n* = 8) and which received a high daily oral dose of FeSO_4_ (60 mg/kg body weight).

Diets were prepared weekly and supplied daily in sufficient quantity to ensure *ad libitum* intake during the 17-day experimental period. Seven grams of oil (buriti or soybean oil) was added for every 100 g of diet. Each rat group was fed with balanced pellet diets prepared according to the American Institute of Nutrition—AIN guidelines [76], as described in the Appendix A.

The soybean oil was added to the control diet since it is the nutritionally recommend lipid source for rodents according to the AIN guidelines [76]. Therefore, the use of another lipid source other than soybean oil, even with a fatty acid composition similar to that of buriti oil, would configure an evaluation of another experimental oil, unlike the proposed study design. Furthermore, soybean oil is one of the most consumed and widely studied oils worldwide and also for this reason it was selected for addition to the control diet [77].

Animal iron overload were induced by gavage of 2 mL of FeSO_4_ (60 mg/kg body weight) was administered by gavage [78], corresponding to approximately 1/5 of lethal dose for rats (LD_50_) [79]. The study design is outlined in Figure 6.

### 4.5. Food Intake, Weight Monitoring and Somatic Parameters

Rat body weights and food intake were evaluated every two days during the 17-day experimental period. Immediately before euthanasia, measurements were conducted as described by Novelli et al. [80] in anesthetized rats. The parameters evaluated were abdominal circumference (AC; immediately anterior to the hind leg), chest circumference (CC; immediately behind the foreleg), body length (BL; measured from their nose to the tail base), and body weight (BW). BL and BW were used to calculate body mass index BMI (BW (g)/length^2^ (cm^2^)) and the Lee index (LI) (cube root of BW (g)/BL (cm)).

### 4.6. Buriti oil Effects on Lipid Profile and Hematological Biochemical Parameters

Rats were anesthetized 24 h after the last treatment by intraperitoneal injection of 1 mL of ketamine hydrochloride (75 mg/kg body weight) associated with 1 mL of xylazine hydrochloride (5 mg/kg body weight), submitted to laparotomy, and euthanized by puncture of the left ventricle. Blood samples were collected by cardiac puncture from each animal to determine the hemoglobin, hematocrit, red blood cells (RBCs), white blood cells (WBCs), platelets, monocytes, leukocytes, and granulocytes, using a cell counter (ABX micro 60, Tokyo, Japan).

Serum was also obtained by centrifugation (1825× *g* for 10 min at 4 °C) to evaluate serum lipids. TAG, TC, HDL, and LDL levels were analyzed using commercial reagent kits according to the manufacturer’s instructions, and absorbance was determined using a LabMax 240 Premium automatic analyzer (Labtest, Belo Horizonte, Brazil) at 500 nm (TC), 600 nm (HDL and LDL), or 505 nm (TG). Very low density lipoprotein cholesterol (VLDL) values were determined using the previously described equation, as follows: VLDL = TG/5 [81].

Aspartate aminotransferase (AST) and alanine aminotransferase (ALT) levels were analyzed to assess hepatic function using commercial Labtest kits (Labtest Diagnóstica S.A.), according to the manufacturer’s instructions.

### 4.7. Antioxidant Activity

Erythrocytes were separated from plasma and then washed three times with 100 mM of potassium phosphate buffer at pH 7.4 and centrifuged (825× *g* at 4 °C for 10 min). These erythrocytes (0.2 mL) were mixed with 1.8 mL of 0.4% β-mercaptoethanol (v/v) and frozen (−20 °C for 20 min). The hemolysate was then centrifuged (8274× *g* at 4 °C for 40 min). Liver tissue samples were homogenized in 50 mM of cold potassium phosphate buffer at pH 7.0, using a Potter–Elvehjem homogenizer to obtain a 10% (w/v) homogenate after centrifugation (8274× *g* at 4 °C for 4 min). 

Both hemolysate and homogenate supernatants were used to determine the GPx (E.C. 1.11.1.9) and SOD (E.C.1.15. 1.1) activities (oxidative stress markers), using commercial kits according to the manufacturer’s instructions. Spectrophotometric measures were performed in triplicate (Shimadzu UV-VIS model 1650-PC spectrophotometer, Tokyo, Japan) and SOD and GPx activities were expressed as IU/mg hemoglobin (Hb) and IU/mg protein for serum and liver, respectively.

### 4.8. Statistical Analysis

A minimum statistical power of 80% was calculated considering the sample size (32 animals divided into four groups, *n* = 8), a minimally detectable effect size of 1.0, and a significance level of 0.05 (α = 0.05). Parametric data were assessed using the Student’s t-test at a significance level of 5% (*p* ≤ 0.05). The results were expressed as mean ± standard deviation (SD). Statistical analyses and graphic design were performed using Sigma Plot 12.5 software for Windows (Systat Software Inc., San Jose, CA, USA). Data pre-treatment with the autoscaling method and calculation of Pearson’s correlation coefficient (r) to measure the association between two variables were carried out in MetaboAnalyst v.5.0 program (Xia Lab, McGill University, Montreal, QC, Canada).

## 5. Conclusions

In summary, this is the first study to indicate the protective effect of in vivo buriti oil consumption against physiological damage induced by an iron overload condition. As a result, buriti oil intake was able to attenuate the deleterious effects caused by the administration of a FeSO_4_ overload. This effect was observed for antioxidant defense, monocytes, and decreased LDL levels, probably due to the chemical composition of the oil, which is rich in unsaturated fatty acids, β-carotene, and α-tocopherol. The results suggest the beneficial effects of buriti oil consumption to maintain redox balance. This oil could be a potential new nutraceutical option to combat pathological disorders associated with iron overload. However, future studies are required to establish its role in controlling status antioxidant and lipid levels, aiming at the development of functional food to treat/improve conditions of athletes or patients experiencing iron overload due to pathological conditions or long-term use.

## Figures and Tables

**Figure 1 molecules-28-02585-f001:**
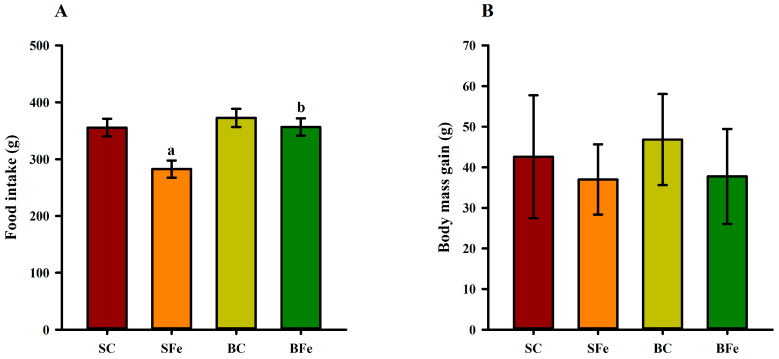
Food intake (**A**) and body mass gain (**B**) of rats treated or not with a high dose of FeSO_4_ and diets added with either soybean oil or buriti oil. SC = control group fed a diet containing soybean oil and gavage with saline solution; BC = control group fed a diet containing buriti oil and gavage with saline solution; SFe = group fed a diet containing soybean oil and gavage with a high-dose FeSO_4_; BFe = group fed a diet containing buriti oil and gavage with a high-dose FeSO_4_. ^a^ Represents a significant difference compared with the SC; ^b^ Represents a significant difference compared with the SFe. Values in the vertical bars are mean ± SD of each rat group (Student’s *t*-test, *p* ≤ 0.05; *n* = 8 rats/group).

**Figure 2 molecules-28-02585-f002:**
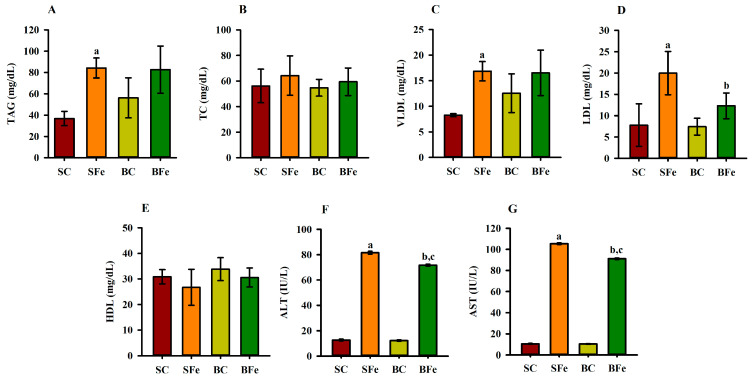
Lipid profile and aminotransferases of rats treated or not with a high dose of FeSO_4_ and diets added with either soybean oil or buriti oil. Triacylglycerides (**A**), total cholesterol (**B**), very low density lipoprotein (**C**), low-density lipoprotein (**D**), high-density lipoprotein (**E**), alanine aminotransferase (**F**), and aspartate aminotransferase (**G**). SC = control group fed diet containing soybean oil and gavage with saline solution; BC = control group fed a diet containing buriti oil and gavage with saline solution; SFe = group fed a diet containing soybean oil and gavage with a high-dose FeSO_4_; BFe = group fed a diet containing buriti oil and gavage with a high-dose FeSO_4_. HDL = high-density lipoprotein; LDL = low-density lipoprotein; TAG = triacylglycerides; TC = total cholesterol; VLDL = very low density lipoprotein. ^a^ Represents a significant difference compared with the SC; ^b^ Represents a significant difference compared with the SFe; ^c^ Represents a significant difference compared with the BC. Values in the vertical bars are mean ± SD of each rat group (Student’s *t*-test, *p* ≤ 0.05; *n* = 8 rats/group).

**Figure 3 molecules-28-02585-f003:**
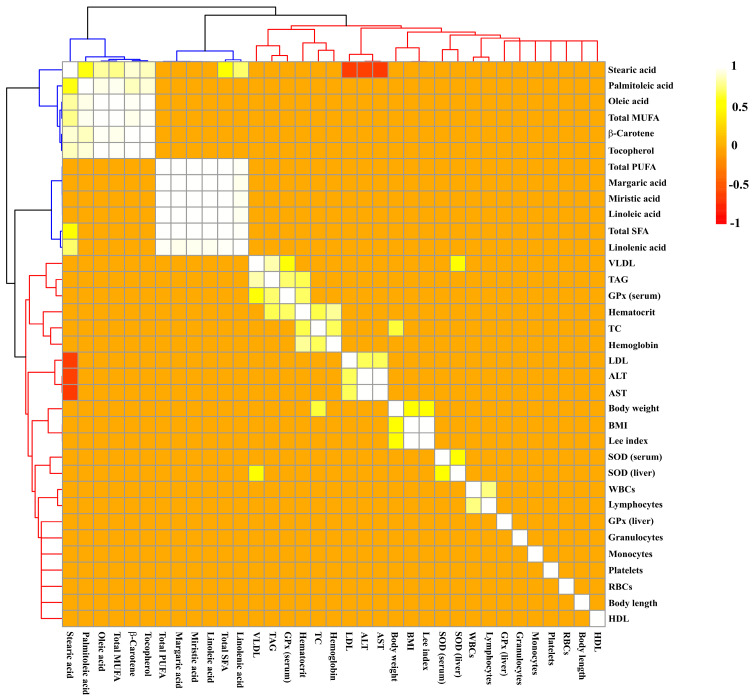
Hierarchical grouping based on Pearson’s correlation matrix between chemical compounds of oils and biological parameters of rats treated or not with a high dose of FeSO_4_ and diets added with either soybean oil or buriti oil. Significant correlations were determined based on r ≥ 0.6 and *p* ≤ 0.05. Positive and negative correlations are shown as yellow-white and red, respectively. Clustering of the correlation matrix reveals two clusters represented by colors: blue (compounds of oils) and red (biological parameters). ALT = alanine aminotransferase; AST = aspartate aminotransferase; BMI = body mass index; DPPH = 1,1-diphenyl-2-picrylhydrazyl; GPx = glutathione peroxidase; HDL = high-density lipoprotein; LDL = low-density lipoprotein; MUFA = monounsaturated fatty acids; PUFA = polyunsaturated fatty acids; RBCs = red blood cells; SFA = saturated fatty acids; SOD = superoxide dismutase; TAG = triacylglycerides; TC = total cholesterol; VLDL = very low density lipoprotein; WBCs = white blood cells.

**Figure 4 molecules-28-02585-f004:**
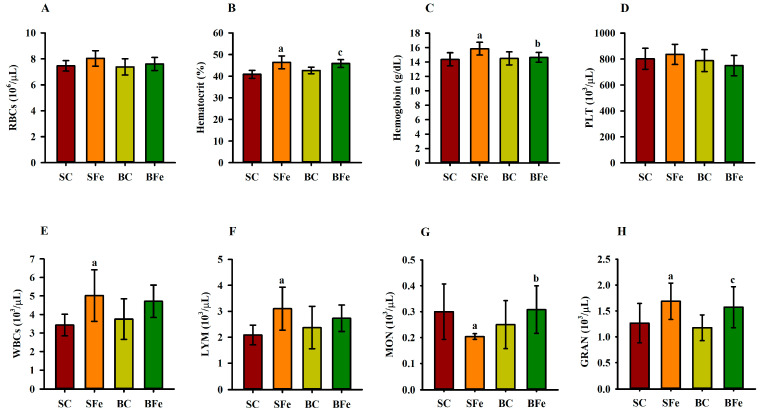
Hematological parameters of rats treated or not with a high dose of FeSO_4_ and diets added with either soybean oil or buriti oil. Red blood cell (**A**), hematocrit (**B**), hemoglobin (**C**), platelet (**D**), white blood cell (**E**), lymphocyte (**F**), monocyte (**G**), and granulocyte (**H**) counts. SC = control group fed a diet containing soybean oil and gavage with saline solution; BC = control group fed a diet containing buriti oil and gavage with saline solution; SFe = group fed a diet containing soybean oil and gavage with a high-dose FeSO_4_; BFe = group fed a diet containing buriti oil and gavage with a high-dose FeSO_4_. RBCs = red blood cells; PLT = platelets; WBCs = white blood cells; LYM = lymphocytes; MON = monocytes; GRAN = granulocytes. ^a^ Represents a significant difference compared with the SC; ^b^ Represents a significant difference compared with the SFe; ^c^ Represents a significant difference compared with the BC. Values in the vertical bars are mean ± SD of each rat group (Student’s *t*-test, *p* ≤ 0.05; *n* = 8 rats/group).

**Figure 5 molecules-28-02585-f005:**
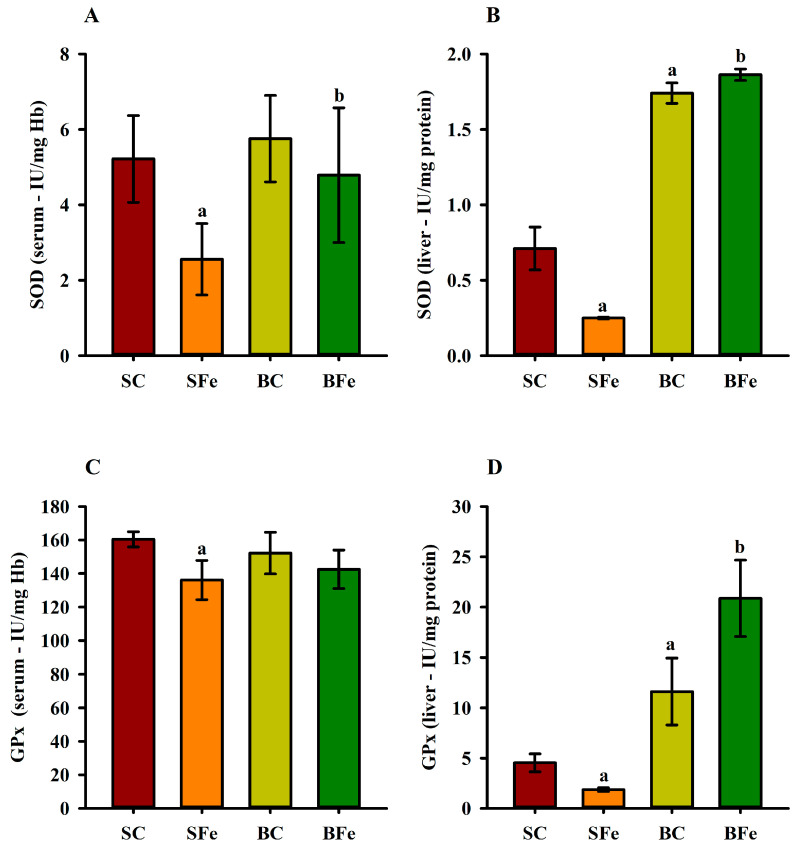
Serum and hepatic superoxide dismutase (**A**,**B**) and glutathione peroxidase (**C**,**D**) activities in rats treated or not with a high dose of FeSO_4_ and diets added with either soybean oil or buriti oil. GPx = glutathione peroxidase; SOD = superoxide dismutase. ^a^ Represents a significant difference compared with the SC; ^b^ Represents a significant difference compared with the SFe. Values in the vertical bars are mean ± SD of each rat group (Student’s *t*-test, *p* ≤ 0.05; *n* = 8 rats/group).

**Figure 6 molecules-28-02585-f006:**
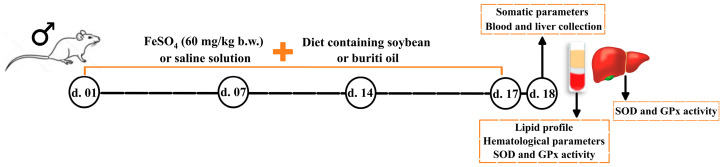
Timeline and steps of study design. B.W. = body weight; d. = day; GPx = glutathione peroxidase; SOD = superoxide dismutase.

**Table 1 molecules-28-02585-t001:** Fatty acids, β-carotene, and α-tocopherol contents in buriti and soybean oils.

Parameters	Soybean Oil	Buriti Oil	*p*
Antioxidant compounds (mg/kg)			
β-carotene	417.07 ± 1.75	787.05 ± 1.55 ^a^	≤0.001
α-tocopherol	302± 1.98	689.02 ± 2.11	≤0.001
Fatty acids (g/100 g)			
Miristic acid—C14:0	16.95 ± 0.04 ^a^	0.70 ± 0.01	≤0.001
Margaric acid—C17:0	10.89 ± 0.30 ^a^	0.20 ± 0.01	≤0.001
Stearic acid—C18:0	3.04 ± 0.20	3.30 ± 0.03	0.09
Total SFA	30.88 ± 0.90 ^a^	4.20 ± 0.05	≤0.001
Palmitoleic acid—C16:1	–	19.0 ± 0.57	
Oleic acid—C18:1	24.02 ± 0.55	72.30 ± 0.80 ^a^	≤0.001
Total MUFA	24.02 ± 0.55	91.30 ± 1.37 ^a^	≤0.001
Linoleic acid—C18:2	40.00 ± 0.15 ^a^	2.40 ± 0.09	≤0.001
Linolenic acid—C18:3	5.01 ± 0.15 ^a^	1.60 ± 0.03	≤0.001
Total PUFA	45.01 ± 0.30 ^a^	4.00 ± 1.20	≤0.001

MUFA = monounsaturated fatty acids; PUFA = polyunsaturated fatty acids; SFA = saturated fatty acids. ^a^ Represent a significant difference between the means ± SD in the same row (Student’s *t*-test, *p* ≤ 0.05).

**Table 2 molecules-28-02585-t002:** Food intake and somatic parameters of rat treated or not with a high dose of FeSO_4_ and diets added with either soybean oil or buriti oil.

Parameters	SC	SFe	BC	BFe	*p*
FBW (g)	369.2 ± 32.5	395.3 ± 27.3	404.7 ± 27.5	387.0 ± 21.4	0.308
FBL (cm)	33.1 ± 1.5	33.3 ± 1.1	33.0 ± 1.9	33.8 ± 1.7	0.774
BMI (g/cm^2^)	0.33 ± <0.1	0.34 ± <0.1	0.36 ± <0.1	0.34 ± <0.1	0.314
Lee index	0.22 ± <0.1	0.22 ± <0.1	0.22 ± <0.1	0.22 ± <0.1	0.586
CC (cm)	15.1 ± 1.0	15.7 ± 0.9	15.9 ± 1.0	15.3 ± 0.9	0.336
AC (cm)	16.8 ± 1.0	16.9 ± 0.6	17.2 ± 1.1	17.0 ± 1.2	0.826
AC/CC ratio	1.1 ± 1.0	1.0 ± 0.7	1.0 ± 1.0	1.1 ± 1.1	0.190

SC = control group fed a diet containing soybean oil and gavage with saline solution; BC = control group fed a diet containing buriti oil and gavage with saline solution; SFe = group fed a diet containing soybean oil and gavage with a high-dose FeSO_4_; BFe = group fed a diet containing buriti oil and gavage with a high-dose FeSO_4_. FBW = final body weight; FBL = final body length; BMI = body mass index; CC = chest circumference; AC = abdominal circumference.

## Data Availability

Not applicable.

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
