# Peer review of "Antioxidant and Lipid-Lowering Effects of Buriti Oil (Mauritia flexuosa L.) Administered to Iron-Overloaded Rats"

_molecules, 2023, doi:10.3390/molecules28062585_

Round 1

Reviewer 1 Report

The active components and functions of Buriti Oil were evaluated. However, the critical molecular mechanisms required more investigation. 

(1) Authors need to perform the pathological examination of target organ.

(2) Whether  Buriti Oil  interacts directly with iron? or inhibited FeSO4 absorption or metabolism?

(3) I also suggested authors need assess the liver function, i.e. test the levels of AST and ALT. 

Author Response

Responses to Reviewer #1:

The active components and functions of Buriti Oil were evaluated. However, the critical molecular mechanisms required more investigation.

Answer: We appreciate your comments/suggestions about the manuscript, which were responded point-by-point. The requested adjustments are included in the revised manuscript. Explanations regarding comments and suggestions are described below.

Comments

  • Authors need to perform the pathological examination of target organ.

Answer: We appreciate the suggestion and agree with the Reviewer on the importance of pathological analysis to fully assess target organ health in model animals, specifically the liver. This analysis was not performed due to infrastructure and budget constraints. Based on the reviewer's comment, we include our data regarding the Alt and AST transaminase analysis in the revised manuscript to resolve the lack of histological examination. Hence, ALT and AST data were included in M&M, Results and Discussion. Please see lines 121 to 122 and 157 to 160 in results, lines 253 to260 in the Discussion, and lines 511 to 513 in Material and Methods.

We understand that not being ideal, but given the limitations described above, the measurement of ALT and AST levels can provide relevant information on liver health. Therefore, based on these results, it is possible to show that no liver damage (light, moderate or severe) was observed in animals used in the present experiment since AST and Alt are widely used as markers of liver injury, providing valuable information on the diagnosis and monitoring of liver diseases. The levels of these transaminases are widely used as markers for liver injury and inflammation in clinical practice by their availability, whose analysis is cheap and has high sensitivity. In short, as biochemical markers, both ALT and AST can provide valuable information on liver health, while histopathological analysis is recommended as a broader evaluation. Budget constraints precluded histopathological analysis, so relying on biochemical markers was the only available option. We understand that it is not ideal, but overall, this enzyme analysis answers the main question/hypothesis of the study. Articles discuss the value of AST and ALT data as surrogate markers for analyzing liver injury and inflammation in clinical practice.

  • Whether Buriti Oil interacts directly with iron? or inhibited FeSO4 absorption or metabolism?

Answer: We agree with the Reviewer regarding the mode of action of the oil. No direct parameter of iron absorption or metabolism was measured since the aim of this study was to assess the buriti oil intake effect in iron-overloaded rats by evaluating somatic parameters, lipid profile, hematological profile and antioxidants due to buriti oil intake in rats with iron overload. Nevertheless, evidence indicates that both β-carotene and oleic acid interfere with iron absorption and metabolism. β-carotene and oleic acid are the major components present in buriti oil, whose chemical composition is very complex. Thus, the high content of β-carotene and oleic acid associated with the diversity of bioactive compounds contained in the oil may explain the attenuation of the absorption and metabolism in animals treated with FeSO4. Therefore, this chemical complexity of the oil may be responsible for a synergistic action, which may explain the observed results. Based on this, we reworked some text in the Discussion, including a possible explanation for responding to the Reviewer's comment. Please see lines 273 to 285 in the revised manuscript.

  • I also suggested authors need assess the liver function, e. test the levels of AST and ALT.

Answer: The Reviewer's observation is pertinent and we agree. ALT and AST results were included as described in Results section, item 2.3 in the revised manuscript. Please see the text and Figure 2 and consequently Figure 3 (referring to correlations). Please see the new Figure 2, the Results section (lines 121 to 123) and the Discussion section (lines 253-260).

Reviewer 2 Report

I recognize that you have worked hard on your research.

This study drew various research results on an interesting topic.

I think that by revising some parts of your research, we can provide the reader with clearer results.

I hope you see my comments and review yours.

1. Refined soybean oil may show differences in the content of beta-carotine or tocopherol depending on the refining process. In addition, there is a heating process in the manufacturing process of buriti oil, so if there is a content to check whether fatty acids are oxidized, it can be meaningful. A more detailed description of the possible process is recommended.

2. In figure 1, it is suggested to change the order of the results in the graph because it is the result of confirming the change of SC to SFe and the change of BC to BFe. (No need to change) The result that there is a significant difference between BC and SFe is meaningless. Rather, it is more meaningful to express that there is no significant difference between SC and BC.

3. In the explanation of Figure 1, food intake is not a one-way anova of four results, so the statistics must be changed.

4. If you are going to explain the abbreviations used in table 2, do them in order. You describe in order of BW, BL, BMI, Lee index---, and in the explanation below, AC, BL, BMI in order.

5. When the body weight in table 2 and the body mass gain value presented in figure 1 were simultaneously checked, it is questionable whether the weight distribution was not appropriate when classifying the experimental animals into groups at the beginning of the experiment. By simple calculation, the average weight of the SC group at the start of the experiment is about 280g, but the BC group is about 327g, the SFe is about 290g, and the BFe group is about 330g.

6. Since it is a large deviation at the beginning of the experiment, problems may occur in all experimental results. Please describe the results of the grouping step.

7. In figure 2, the author also tested whether there was a significant difference between SC and SFe, and whether there was a significant difference between BC and BFe, respectively. Comparisons between two items do not use one-way ANOVA. (using t-test)

8. In figure 3, the correlation was confirmed, but it is necessary to check whether the p-value of each value is significant.

9. As a result of figure 4, the effect of restoring the normal state of buriti oil compared to soybean oil when excessive Fe intake is statistically unique to Hemoglobin. In addition, SC and BC are not in a healthy state (normal state) when viewed as an experimental method, so comparison with a negative control (a group that does not consume Fe or oil) is appropriate. Provide explanations of abbreviations in order of limiting figures.

10. In the case of figure 5, the difference in hepatic antioxidant enzyme activity compared to serum was significant. The reason is not even in the discussion. I need an explanation.

Author Response

Responses to Reviewer #2:

Comments

I recognize that you have worked hard on your research. This study drew various research results on an interesting topic. I think that by revising some parts of your research, we can provide the reader with clearer results. I hope you see my comments and review yours.

Answer: We thank you for acknowledging the merit of our study. Also, we thank you for all your comments on the manuscript, which were answered point-by-point. Requested adjustments are included in the revised manuscript, while explanations for comments are described below.

  1. Refined soybean oil may show differences in the content of beta-carotene or tocopherol depending on the refining process. In addition, there is a heating process in the manufacturing process of buriti oil, so if there is a content to check whether fatty acids are oxidized, it can be meaningful. A more detailed description of the possible process is recommended.

Answer: We agree with the Reviewer. Previous studies demonstrated that the refining process reduces the carotenoid and tocopherol content in soybean and buriti oils, despite improving thermal and oxidative stability and quality. For this reason, although buriti oil extraction involves mild heating (60°C/30 min), after the refinement process, the buriti oil administered to BC and BFe rat groups was previous analysed from a nutritional point of view (Table 1 - Results Section) and as well as the variables: free fatty acids (0.29±0.03 %), acidity index (0.22±0.03 %) and peroxide index (6.89±0.78 mEq/kg). This analysis demonstrated that the composition of the refined buriti oil complies with the normal standards for consumption (Aquino et al., 2012). These analyses are important to guarantee the quality of the oil administered to the rats. We have included this information in the Materials and Methods Section. Please see lines 394 to 407.

Reference cited above: Aquino, J.d.S.; Pessoa, D.C.N.d.P.; Araújo, K.d.L.G.V.; Epaminondas, P.S.; Schuler, A.R.P.; Souza, A.G.d.; Stamford, T.L.M. Refining of buriti oil (Mauritia flexuosa) originated from the Brazilian Cerrado: physicochemical, thermal-oxidative and nutritional implications. J. Braz. Chem. Soc. 2012, 23, 212-219.

  1. In figure 1, it is suggested to change the order of the results in the graph because it is the result of confirming the change of SC to SFe and the change of BC to BFe. (No need to change) The result that there is a significant difference between BC and SFe is meaningless. Rather, it is more meaningful to express that there is no significant difference between SC and BC.

Answer: As requested by the Reviewer, we modified the order of presentation of the group results both in the Tables and in the Figures, excluding the statistical difference between BC and SFe. In the new version, the sequence is SC, SFe, BC and BFe. We also checked the statistics on all results and changed the captions of all Tables and Figures. Please see the new version of the Results Section in the revised manuscript version.

  1. In the explanation of Figure 1, food intake is not a one-way anova of four results, so the statistics must be changed.

Answer: Thanks for the reviewer's remarks. Data from the four groups can be compared either by one-way ANOVA (single factor for the four-level group) followed by Tukey's Post hoc test for multiple comparisons or compared two by two by Student's t-test. However, we chose to use the Student's t-test to compare data from SC versus SFe, BC versus BFe, SC versus BC and SFe versus BFe.

  1. If you are going to explain the abbreviations used in table 2, do them in order. You describe in order of BW, BL, BMI, Lee index---, and in the explanation below, AC, BL, BMI in order.

Answer: We change the caption of Table 2, placing the parameters in the order in the caption.

  1. When the body weight in table 2 and the body mass gain value presented in figure 1 were simultaneously checked, it is questionable whether the weight distribution was not appropriate when classifying the experimental animals into groups at the beginning of the experiment. By simple calculation, the average weight of the SC group at the beginning of the experiment is about 280g, but the BC group is about 327g, the SFe is about 290g, and the BFe group is about 330g.

Answer: All rats approximately 80 days old at the start of the experiment were randomly distributed in the groups, according the initial weight. After reviewing the data, we found a typo in the table. It is corrected in the revised manuscript version. We recalculated the statistics presented in Figure 1B and Table 2. We apologize for this mistake.

  1. Since it is a large deviation at the beginning of the experiment, problems may occur in all experimental results. Please describe the results of the grouping step.

Answer: On the first day of the experiment, the rats were randomized into four groups, with initial weights of 326.60 ± 28.84 g for SC, 358.30 ± 31.79 g for SFe, 360.50 ± 12.97 g for BC, and 349.30 ± 33.38 g for BFe, without statistical differences by Student's t-test. Please see the new version of Figure 1 (weight gain) and the final weights of the rats in the experiment shown in Table 2.

  1. In figure 2, the author also tested whether there was a significant difference between SC and SFe, and whether there was a significant difference between BC and BFe, respectively. Comparisons between two items do not use one-way ANOVA. (using t-test)

Answer: We appreciate the reviewer's remarks and recognize that it is no necessary to include statistical comparisons between BC and SFe. Thus, the Student's t-test was used for comparison between two groups instead of the one-way ANOVA (single factor for the four-level group) followed by Tukey's Post hoc test for multiple comparisons. Please see Figure 2 and its caption.

  1. In figure 3, the correlation was confirmed, but it is necessary to check whether the p-value of each value is significant.

Answer: We included the p-value in Table S2 (Supplementary Material) as requested by the Reviewer. Its direct insertion in Figure 3 could hinder its visualization.

  1. As a result of figure 4, the effect of restoring the normal state of buriti oil compared to soybean oil when excessive Fe intake is statistically unique to Hemoglobin. In addition, SC and BC are not in a healthy state (normal state) when viewed as an experimental method, so comparison with a negative control (a group that does not consume Fe or oil) is appropriate. Provide explanations of abbreviations in order of limiting figures.

Answer: We appreciate the reviewer's comment. Nevertheless, the effect of restoring the normal state of BFe compared to SFe (iron overloaded rats) is statistically significant on hemoglobin and monocytes (Please see Figure 4 and it caption).

Regarding the SC group, it was correctly considered as a control group and a healthy group since it was not treated with oral ferrous sulfate and its diet followed the American Institute of Nutrition (AIN) recommendations. The AIN advocates soybean oil as a fat source in the diet of rats (Reeves et al., 1993), regardless of the type of study, as it is a guideline for rodents. For this reason, we did not consider it necessary to include a group treated without both oil and iron since we have the SC group.

In addition, the group with a diet added with buriti oil (BC) presents normal status since no disease was induced. There was only one change in the lipid source of the diet (oil) as recommended by the AIN-93 (Reeves et al., 1993), and both diets have the same lipid percentage. In a previous study, Aquino et al. (2015) showed a state of normality in rats that consumed these same diets (with the addition of soy or buriti oil) for 28 days, that is, for a longer period than that used in the present study., which was 17 days.

References cited above:

Reeves, P.G.; Nielsen, F.H.; Fahey Jr, G.C. AIN-93 purified diets for laboratory rodents: final report of the American Institute of Nutrition ad hoc writing committee on the reformulation of the AIN-76A rodent diet. J. Nutr. 1993, 123, 1939-1951.

Aquino, J.S.; Soares, J.K.B.; Magnani, M.; Stamford, T.C.M.; Mascarenhas, R.D.J.; Tavares, R.L.; Stamford, T.L.M. Effects of dietary Brazilian palm oil (Mauritia flexuosa L.) on cholesterol profile and vitamin A and E status of rats. Molecules 2015, 20, 9054-9070.

  1. In the case of figure 5, the difference in hepatic antioxidant enzyme activity compared to serum was significant. The reason is not even in the discussion. I need an explanation.

Answer: As requested by the reviewer, we have included an explanation in the discussion regarding the significant differences in serum liver antioxidant enzyme activity. Please see lines 334 to 351 in the revised manuscript. This difference can be explained due to the early effect caused by changes in diet and iron overload in the liver compared to the effect provoked at the systemic level. Antioxidant enzymes patterns in rats fed different dietary lipid sources are tissue-specific, which may modify responses to oxidative stress in specific organs. Regarding the liver, data suggest a particular organ vulnerability and the importance of considering the tissue-specific differences to assess antioxidant enzyme activity during treatments.

Round 2

Reviewer 2 Report

Dear author, 

When I checked your response to the inquiry about your article, I confirmed that you gave an appropriate and detailed answer and corrected some of the contents.

I think your article is suitable for publication in a journal.

I'm happy to review a good subject.

I hope you have a nice day.